# What are the volume and budget needs to provide chemotherapy to all children with acute lymphoblastic leukaemia in Thailand? Development and application of an estimation tool

Rosarin Sruamsiri [1], Alessandra Ferrario [2], Dennis Ross-Degnan [2], Avram E Denburg [3,4,5,6], A Lindsay Frazier [7], Sumit Gupta [3,4,5,6], Zachary J Ward [8], Jennifer M Yeh [9,10], Anita Katharina Wagner [2]

RS and AF are joint first authors.

For numbered affiliations see end of article.

**Correspondence to**
Dr Alessandra Ferrario;
alessandra_ferrario@hphci.harvard.edu

## ABSTRACT

**Objective** Insufficient access to anticancer medicines may contribute to the wide survival differences of children with cancers across the globe. We developed a tool to estimate the volume of medicines and budget requirements to provide chemotherapy to children with acute lymphoblastic leukaemia (ALL).

**Design** Development and application of an estimation tool.

**Setting** Paediatric oncology hospital departments in Thailand.

**Participants** 318 children aged 0–14 years diagnosed with ALL and 215 children with undiagnosed ALL.

**Interventions** Estimates of volume and budget requirements for administering a full course of chemotherapy for ALL and a further course for children who relapse, according to National Treatment Guidelines.

**Primary and secondary outcome measures** Primary outcome measures were the volume (mg) and cost (US$) of medicines needed to treat children with ALL. For medicines whose main indication is paediatric ALL (asparaginase and 6-mercaptopurine), we estimated the difference between volume needed and actual sales in 2017 (secondary outcome).

**Results** Ten anticancer medicines and four chemoprotective agents are needed for the treatment of paediatric ALL according to the Thai treatment guidelines. Of these 14 medicines, 13 are included in the WHO essential medicines list for children. All are available as generics. We estimated that essential chemotherapy and chemoprotective agents to treat all children diagnosed with ALL in Thailand in 2017 would cost US$ 814 952 (US$ 1 365 422 for diagnosed and undiagnosed children), which corresponds to 0.005% (0.008%) of the country's total health expenditure. The volumes of asparaginase and 6-mercaptopurine available on the Thai market in 2017 were more than sufficient (2.3 and 1.5 times the amounts needed, respectively) to treat all children diagnosed with ALL.

**Conclusions** Procuring sufficient quantities of essential medicines to treat children with ALL requires relatively modest resources. Medicine cost should not be a major barrier to ALL treatment in similar settings.

## Strengths and limitations of this study

► The tool is dynamic and facilitates transparency for estimating the volume of medicines and budget requirements.

► It allows users to input local data of disease incidence, treatment protocols and medicine prices.

► Our application uses data on cancer incidence, prices of medicines and the volume of medicines available in Thailand.

► While informed by local data, we had to make some assumptions on cancer stage distributions.

## BACKGROUND

Important global and local forces have shaped the paediatric oncology landscape over the past two decades. The paediatric oncology community, together with researchers and other partners, has advocated widely for the need to address disparities in survival between countries at different levels of health system development.[1–3] Global efforts to close this gap have featured the development of resource-appropriate treatment guidelines,[4,5] the inclusion of key medicines to treat paediatric cancer in iterative updates of the WHO's Essential Medicines List for Children,[6] the establishment of twinning programmes[7,8] and global working groups like the WHO Global Initiative for Childhood Cancer,[9] the development and strengthening of paediatric cancer registries and analytical efforts to provide evidence on the burden of paediatric cancer.[10–14] These paediatric oncology focused endeavours have been supported by broader global commitments and efforts to achieve universal health coverage and the sustainable development agenda, which, among other targets, emphasises reduction in

preventable mortality from non-communicable diseases and access to quality-assured essential medicines.[15 16]

At the country level, efforts have included the development and implementation of standardised protocols to treat paediatric cancers, the establishment of national cancer plans and cancer registries and the addition of paediatric cancer medicines to national essential medicines and reimbursement lists,[17] among others. The extent to which different countries have engaged in such efforts varies. Thailand has been in the vanguard of these positive global trends. In Thailand, national treatment guidelines for childhood leukaemia were developed in 2005 and implemented in 2006, together with coverage for diagnostics, chemotherapy and hospital treatment by the National Health Security Office (NHSO).[18–20] Paediatric cancer registries are in place across the country and regular analyses of incidence and survival have been published over time.[18–22] At 67%, 5-year net survival of children with acute lymphoblastic leukaemia (ALL) (the most common form of cancer in children in most regions of the world) in Thailand is higher than the global (56%), Asian (53%), South-Eastern Asian (40%) and the upper middle-income countries averages (63%) but lower than survival in high-income countries (88% on average)[13] (see online supplemental table 1 for more indicators).

A key component of paediatric cancer treatment is chemotherapy. Yet, despite calls for making medicines affordable and the need to address shortages affecting paediatric cancer medicines globally,[1 23 24] analyses of availability of paediatric cancer medicines at the country level are limited. Previous multicountry studies reviewed the inclusion of paediatric cancer medicines in national essential medicines and reimbursement lists[25] and surveyed physician and healthcare professionals regarding the level of access to essential paediatric chemotherapies.[26] Subnational studies have assessed medicines availability and affordability at the pharmacy level.[27] These are important contributions that need to be complemented by evidence on the actual volumes of medicines needed and available, and the corresponding budget required to provide a full course of treatment to all children with cancers. Further, longstanding supply chain vulnerabilities impacting access to quality medicines have been dramatically exposed during the COVID-19 pandemic,[28 29] calling for even greater attention to availability of medicines.

Together with other health system barriers, insufficient access to anticancer medicines may contribute to the wide survival differences of children with cancers across the globe. The objective of our study was to develop a tool to estimate medicine needs and budgetary requirements for treating children with ALL and apply it to Thailand as an example case. We populated the tool with a unique set of data including the latest available estimates of children with diagnosed and undiagnosed ALL, pharmaceutical sales data and local data on cancer stage distribution. Further, we compared needed quantities with sold quantities of paediatric cancer medicines used for the treatment of ALL to identify possible unmet need. Ultimately, we aimed to generate a transferable approach that can be customised to other jurisdictions and populated with local data to estimate the need for and availability of essential paediatric cancer medicines as well as their corresponding budget impact. We expect this tool to be useful for procurement planning, budgeting, advocacy and monitoring of sufficiency of medicine supplies.

## METHODS
### Data sources
#### Cancer incidence by risk group or stage
We used age-stratified (<1; 1–4, 5–9, 10–14 years old) estimates of the number of children with incident ALL, diagnosed and undiagnosed, from the Global Childhood Cancer (GCC) microsimulation model.[12] These GCC estimates were calibrated against the International Incidence of Childhood Cancer data[30] and included relevant demographic, incidence and health system variables to estimate the number of diagnosed and undiagnosed cases.[12] As published international estimates do not subclassify patients by risk group, we estimated the proportion of children with ALL by risk group (standard, high, relapse) using data on the proportion of children per risk group treated with standard versus high-risk ALL protocols in Thailand.[19]

#### Therapeutic guidelines
We used the national treatment guidelines published in 2016 by the Thai NHSO,[31] which were developed using treatment regimens implemented in advanced healthcare systems. NHSO guidelines are the basis for health insurance reimbursement for approximately 80% of the Thai population. The guidelines include separate protocols for standard-risk ALL, high risk ALL and for relapsed ALL (the latter with separate treatment schemes for standard and high-risk relapse). All protocols specify recommended doses of each medication by age group, risk group and treatment phase. We identified all antineoplastic agents and chemoprotective agents in the 2016 NHSO paediatric ALL treatment guidelines.[31]

#### Medications sold
We estimated the volumes of each medication of interest sold in Thailand from 1 January 2017 to 31 December 2017 using IQVIA (formerly IMS Health and Quintiles) data. IQVIA is a multinational company that conducts multisample audits of pharmaceutical purchase data based on invoices from pharmacies, wholesalers, distributors and manufacturers in the hospital and retail sectors in a number of countries worldwide.[32] The proprietary data are extrapolated to represent national level sales. Quality checks (eg, comparison with manufacturer data) are conducted to ensure accuracy and representativeness of the data.[32] IQVIA sales data record quarterly sales volumes of each product with information on formulation, strength, number of units per pack and number of

packs sold. We used this information to calculate the total volume, in mg, of each active ingredient sold in 2017. We expressed volume of medicines sold as the number of smallest units for the relevant formulation (ie, the one recommended in the Thai paediatric ALL guidelines) of each international non-proprietary name. We defined the smallest unit as the unit with the smallest amount (mg or active units) of active ingredient for the relevant formulation listed in the 2019 WHO model list of essential medicines for children.[6]

## Prices

We extracted price data from the Pharmaceutical Information Center, Ministry of Public Health, Thailand.[33] These represent list prices and may be higher than the prices actually paid. As such, they provide a conservative estimate of the maximum price paid.

## Analyses

### Paediatric ALL medication need

Building on earlier work (2014) by some of the authors (RS, DR-D, AKW and ALF), we updated a tool that we created to forecast medicine need. We used data on weight and height of Thai children by age[34] to calculate the average body surface area for each age group according to the Mosteller formula.[35] We multiplied the number of patients in each age and risk group by the milligrams (or active units) per body surface area needed, for each ALL medication, according to the treatment protocols. We then summed the number of mg needed across different age and risk groups to arrive at a total number of milligrams for each antineoplastic and chemoprotective agents needed to treat both diagnosed and undiagnosed children with ALL in Thailand. We present the volume of ALL medicines needed as the number of smallest units.

### Comparing volumes needed and volumes sold

Of the 10 anticancer medicines included in the paediatric ALL protocols, most are also used for high-incidence adult cancers like breast cancer and lung cancer. Only two medicines, asparaginase and 6-mercaptopurine, are used predominantly for paediatric ALL, with a relatively small proportion of the total volume used for adolescents and young adults with ALL. In Thailand, asparaginase is a minor agent in treatment protocol for acute myeloid leukaemia (in 2017, there were 97 children diagnosed with this type of cancer in Thailand). For these two medications, we compared the smallest units of both estimated needs and national sales volumes between January and December 2017.

### Expenditure

Using price and volume data, we calculated the budget needed to purchase antineoplastic and chemoprotective agents for diagnosed only and diagnosed and undiagnosed children in 2017 and 2021. Prices were adjusted for inflation.[36] For the 2017 estimates, we calculated what percentage of the national healthcare expenditure[37] was required to pay for these medicines.

We conducted sensitivity analyses by varying the proportion of children who relapse and the protocol on which they relapse (standard vs high risk) and by using the upper bound of the estimate of the number of children with ALL.[12] Further, we tested the additional budgetary requirements if 20% and 40% of each vial content is wasted due to larger than needed vial size. Data preparation was conducted in Excel and the final analysis was conducted in Stata V.15.1 (StataCorp LLC).

This study was approved by the Harvard Pilgrim Health Care Institute Institutional Review Board. The corresponding authors had full access to all data and the final responsibility to submit for publication. We used the STROBE (Strengthening the Reporting of Observational Studies in Epidemiology) guidelines[38] in the writing of this study.

### Patient and public involvement

Patients or the public were not involved in the design, or conduct, or reporting or dissemination plans of our research.

## RESULTS

Based on the GCC microsimulation model, there were an estimated 318 (95% uncertainty interval 197–434) children 0–14 years old with diagnosed ALL and 215 children with undiagnosed ALL in Thailand in 2017, for a total of 533 (95% uncertainty interval 296–795) children with ALL (table 1). Children who develop ALL from age 10–14 are considered high risk, so all children in this age group were allocated to the high-risk protocol. Based on published local evidence of stage distribution and relapse,[19] 65% of 0 to 9-year-old children were categorised as standard risk and 35% as high risk. Further, we estimated that 69 diagnosed children (22%) would relapse (table 2). Of these relapses, we estimated that 28% would be treated on the standard-risk relapse protocol and 72% on the high-risk relapse protocol based on evidence from previous local studies on immunotype and site of relapse,[19 21] risk allocation according to the national treatment guidelines[31] and assuming equal distribution of very early, early and late relapse.

Ten anticancer medicines and four chemoprotective agents are employed in the treatment of paediatric ALL in Thailand. Thirteen of these 14 medicines are included in the WHO essential medicines list for children and all medicines are available as generics. Providing these medicines to all diagnosed children would cost an estimated US\$ 814 952 (US\$ 1 365 422 including undiagnosed children), which represents 0.005% (0.008% including undiagnosed) of total health expenditure in Thailand in 2017 (table 3). Five medicines would account for more than 90% of total expenditure (asparaginase 28%, methotrexate 26%, 6-mercaptopurine 20%, cytarabine 9% and vincristine 9%). For medicines used almost exclusively for the treatment of paediatric ALL, asparaginase and 6-mercaptopurine, the volume sold is more than sufficient

**Table 1** Estimated numbers of Thai children aged 0–14 years with ALL in 2017 and 2021

| | Diagnosed | | | | | | | | Diagnosed and undiagnosed | | | | | | | |
| | Standard risk | | | | High risk | | | | Standard risk | | | | High risk | | | |
| | n | LB | UB | % | n | LB | UB | % | n | LB | UB | % | n | LB | UB | % |
|---|---|---|---|---|---|---|---|---|---|---|---|---|---|---|---|---|
| **2017** | | | | | | | | | | | | | | | | |
| *Age group* | | | | | | | | | | | | | | | | |
| 0–4 | 95 | 58 | 128 | 29.8 | 51 | 32 | 70 | 16.1 | 159 | 87 | 238 | 29.8 | 86 | 47 | 129 | 16.1 |
| 5–9 | 68 | 43 | 92 | 21.3 | 37 | 23 | 50 | 11.5 | 113 | 64 | 169 | 21.3 | 61 | 34 | 92 | 11.5 |
| 10–14 | 0 | 0 | 0 | 0 | 68 | 41 | 94 | 21.3 | 0 | 0 | 0 | 0.0 | 113 | 64 | 167 | 21.2 |
| Total by risk group | 162 | 101 | 221 | 51.1 | 155 | 96 | 213 | 48.9 | 272 | 151 | 408 | 51.1 | 261 | 145 | 387 | 48.9 |
| Total by diagnosis status | 318 | 197 | 434 | 100 | | | | | 533 | 296 | 795 | 100 | | | | |
| **2021** | | | | | | | | | | | | | | | | |
| 0–4 | 86 | 51 | 120 | 29.0 | 47 | 28 | 65 | 15.7 | 144 | 78 | 219 | 29.0 | 78 | 42 | 118 | 15.7 |
| 5–9 | 64 | 42 | 90 | 21.6 | 35 | 22 | 48 | 11.7 | 108 | 62 | 163 | 21.8 | 59 | 33 | 88 | 11.8 |
| 10–14 | 0 | 0 | 0 | 0.0 | 65 | 40 | 91 | 21.9 | 0 | 0 | 0 | 0.0 | 108 | 59 | 161 | 21.7 |
| Total by risk group | 150 | 93 | 210 | 50.7 | 146 | 90 | 204 | 49.3 | 252 | 140 | 382 | 50.8 | 244 | 134 | 367 | 49.2 |
| Total by diagnosis status | 297 | 183 | 414 | 100 | | | | | 496 | 274 | 749 | 100 | | | | |

LB and UB of 95% uncertainty interval (calculated as the 2.5 and 97.5 percentiles). Numbers (n, LB and UB) were rounded to the nearest full digit.
Sources: Number of incident paediatric ALL cases (diagnosed, all) by age group[12], risk group proportions for children.[19]
ALL, acute lymphoblastic leukaemia; LB, lower bound; UB, upper bound.

(2.3 and 1.5 times, respectively, the amounts needed) to treat all children diagnosed with ALL, including children who relapse.

Sensitivity analyses showed that even if the number of children with paediatric ALL is 27% higher (upper bound of 95% uncertainty interval), or if 50% of children relapse or if the ratio of standard-risk versus high-risk relapse is 60% greater, the total expenditure required to treat diagnosed and undiagnosed children would still be less than 0.02% of total health expenditure (see online supplemental material). If 20% of the vial content (for all medicines that listed in the protocols to be administered in vials) is wasted, an additional US$ 123 439 would be required to treat all diagnosed children in 2017. The amount increases to US$ 246 876 if 40% of the volume in each vial is wasted.

**Table 2** Estimated numbers of Thai children with ALL who relapse by risk group in 2017 and 2021

| | Diagnosed | | | | | | | | Diagnosed and undiagnosed | | | | | | | |
| | Standard risk | | | | High risk | | | | Standard risk | | | | High risk | | | |
| | n | LB | UB | % | n | LB | UB | % | n | LB | UB | % | n | LB | UB | % |
|---|---|---|---|---|---|---|---|---|---|---|---|---|---|---|---|---|
| **2017** | | | | | | | | | | | | | | | | |
| Total relapse by risk group* | 19 | 12 | 26 | 28.0 | 50 | 31 | 68 | 72.0 | 32 | 18 | 48 | 28.0 | 83 | 46 | 124 | 72.0 |
| Total relapse by diagnosis status | 69 | 43 | 94 | 100 | | | | | 116 | 64 | 172 | 100 | | | | |
| **2021** | | | | | | | | | | | | | | | | |
| Total relapse by risk group* | 18 | 11 | 25 | 28.0 | 46 | 29 | 65 | 72.0 | 30 | 17 | 46 | 28.0 | 78 | 43 | 117 | 72.0 |
| Total relapse by diagnosis status | 64 | 40 | 90 | 100 | | | | | 108 | 59 | 162 | 100 | | | | |

LB and UB of 95% uncertainty interval (calculated as the 2.5 and 97.5 percentiles). Numbers (n, LB and UB) were rounded to the nearest full digit.
*Risk classification for relapse treatment is independent of the original risk classification for the first full course of treatment. There is a second risk assessment for children who relapse.
Sources: Number of incident paediatric ALL cases (diagnosed, all) by age group,[12] risk group proportions for children.[19 21]
ALL, acute lymphoblastic leukaemia; LB, lower bound; UB, upper bound.

**Table 3** Medications for paediatric ALL—estimated volumes needed, cost and actual sales in Thailand in 2017 and 2021

| | Route | Smallest unit | 2017 Total smallest units | | 2017 Expenditure (US$) | | Percentage | 2021 Total smallest units | | 2021 Expenditure (US$) | |
|---|---|---|---|---|---|---|---|---|---|---|---|
| | | | DX | DX+uDX | DX | DX+uDX | | DX | DX+uDX | DX | DX+uDX |
| Asparaginase | IM | 10 000 | 4,607(10,365) | 7719 | 225 847 | 378 394 | 27.7 | 4348 | 7263 | 236 209 | 394 571 |
| Methotrexate | All | | | | 213 031 | 356 925 | 26.1 | | | 222 654 | 371 993 |
| Methotrexate | Intravenous | 50 | 83 180 | 139 364 | 190 358 | 318 934 | | 78 482 | 131 109 | 199 042 | 332 511 |
| Methotrexate | Oral | 2.5 | 235 036 | 393 798 | 19 285 | 32 311 | | 221 110 | 369 701 | 20 105 | 33 617 |
| Methotrexate | IT | 10 | 6730 | 11 281 | 3388 | 5680 | | 6285 | 10 512 | 3507 | 5865 |
| Mercaptopurine | Oral | 50 | 223,501(340,035) | 374 472 | 164 313 | 275 303 | 20.2 | 210 281 | 351 584 | 171 324 | 286 448 |
| Cytarabine | Intravenous | 100 | 18 905 | 31 674 | 73 572 | 123 268 | 9.0 | 17 798 | 29 752 | 76 763 | 128 319 |
| Vincristine | Intravenous | 1 | 17 363 | 29 092 | 70 469 | 118 070 | 8.6 | 16 341 | 27 320 | 73 507 | 122 891 |
| Tioguanine | Oral | 40 | 5429 | 9096 | 12 011 | 20 125 | 1.5 | 5107 | 8539 | 12 522 | 20 938 |
| Mitoxantrone | Intravenous | 10 | 112 | 188 | 11 586 | 19 413 | 1.4 | 105 | 176 | 12 081 | 20 199 |
| Doxorubicin | Intravenous | 10 | 3401 | 5698 | 9705 | 16 260 | 1.2 | 3211 | 5363 | 10 155 | 16 961 |
| Cyclophosphamide | Intravenous | 500 | 1386 | 2322 | 6551 | 10 976 | 0.8 | 1309 | 2186 | 6854 | 11 448 |
| Etoposide | Intravenous | 100 | 375 | 628 | 2112 | 3538 | 0.3 | | 589 | 2202 | 3681 |
| Leucovorin | Intravenous | 30 | 3438 | 5761 | 21 030 | 35 236 | 2.6 | 3235 | 5409 | 21 928 | 36 663 |
| MESNA | Intravenous | 400 | 822 | 1378 | 2565 | 4298 | 0.3 | 780 | 1301 | 2401 | 4004 |
| Prednisolone | Oral | 5 | 183 024 | 306 654 | 1863 | 3122 | 0.2 | 172 180 | 287 889 | 1943 | 3249 |
| Dexamethasone | Oral | 2 | 21 048 | 35 265 | 295 | 494 | 0.0 | 19 801 | 33 107 | 307 | 514 |
| Antineoplastic agents | | | | | 789 198 | 1 322 272 | | | | 824 270 | 1 377 449 |
| Chemoprotective agents | | | | | 25 754 | 43 149 | | | | 26 579 | 44 430 |
| Total expenditure on antineoplastic and chemoprotective agents | | | | | 814 952 | 1 365 422 | | | | 850 849 | 1 421 880 |
| Total health expenditure (in million constant (2017) US$) | | | | | 17 055 | | | | | | |
| Percentage of total health budget | | | | | 0.005 | 0.008 | | | | | |

Route of administration: (volume sold) is provided for medicines used mainly for the treatment of paediatric ALL. Tioguanine is also known as thioguanine.

Source: Authors' estimation based on Thai ALL treatment protocol,[31] incidence data,[12] IQVIA sales data, WHO Global Health Expenditure Database.[37]

ALL, acute lymphoblastic leukaemia; DX, diagnosed; DX+uDX, diagnosed and undiagnosed; IM, intramuscular; IT, intrathecal.

## DISCUSSION

In this study, we contribute to global and local efforts to improve paediatric cancer treatment with the development of a tool to estimate the volume of medicines needed and available and the corresponding budget required to implement national treatment protocols for paediatric ALL. We find that providing essential medicines to all children with ALL (diagnosed and undiagnosed) in Thailand would cost US$ 1 365 422, which corresponds to less than 0.01% of the country's 2017 healthcare expenditure. To help inform procurement planning going forward, we also estimated volume and budgetary requirements for 2021 and found that accounting for inflation and changes in incidence, the expected budget impact of paediatric ALL antineoplastic and chemoprotective medications is US$ 1 421 880 (diagnosed and undiagnosed). Treating the most common form of paediatric cancer thus requires a relatively modest investment. The strengths of our tool are that it is based on local treatment protocols, local incidence estimates of both diagnosed and undiagnosed children and local medicines prices and sales; the model can readily be customised for use by other countries who populate it with their own data.

Access to essential paediatric oncology medicines is a long-standing[1] and complex issue. Certain barriers to access disproportionately impact countries with developing health systems, such as limited affordability, inefficient procurement practices, supply chain weaknesses and poor quality medicines.[39] However, countries at all levels of health system development are affected by recurrent shortages of essential paediatric antineoplastic medicines.[40] These shortages are due to a range of factors including thin profit margins for generic antineoplastics, interruptions in sole source production due to issues in the manufacture of sterile injectables and poor forecasting of need.[41] Shortages introduce risks of suboptimal treatment due to treatment interruptions, substitution and increased risk of exposure to substandard or falsified medicines procured reactively through available alternative suppliers.[42 43] Availability challenges have been amplified during the COVID-19 pandemic. It is therefore very timely to reinforce the importance of estimating needed procurement volumes and budgets.[44] The tool we developed can support country planners.

A national essential medicines list linked with standard treatment guidelines and universal health coverage have been identified as keys to improved access to paediatric cancer treatment and ultimately health outcomes. Given the imperative of using scarce resources wisely on the path to universal health coverage, it is important to know the extent of treatment needs and expected budget impacts. Accurate forecasts of need can help increase procurement efficiency. If procurement is aggregated across different hospitals, it can generate economies of scale.[44] Knowledge of the types of medicines needed and volumes required can enable measures to proactively prevent or mitigate the impact of anticipated or recurrent shortages.

There are limitations to our study. Our model includes some assumptions. We assume that all children complete the full first course of treatment and that those who relapse completely the full relapse protocol. We made these assumptions because we wanted to estimate the volume and budget that would be needed if every child received a full treatment course. In reality, some children will relapse, die from ALL or treatment toxicity or abandon therapy during their treatment course. The Thai guidelines include separate treatment regimens for low-risk infants and intermediate-risk/high-risk infants. Given that less than 4% of children with ALL in Thailand were infants (<1-year old), we included them in the 0 to 4-year-old group, which was assigned proportions of 0.65 and 0.35 to either the standard-risk or high-risk protocols, respectively. The distributions of risk status and assigned treatment protocol were based on assumptions informed by local data. Sensitivity analyses varying these distributions did not change our overall conclusions. The IQVIA sales data do not capture medications obtained through special procurement channels, such as donated products. We may therefore have underestimated the available volumes of some products. However, as we found no gaps in availability, this would not affect our finding of sufficient amounts of asparaginase and 6-mercaptopurine in Thailand at the time of the study. Finally, while availability of medicines is an essential component of effective paediatric cancer treatment, improving outcomes of care also requires other health systems inputs (eg, paediatric oncologists and nurses) and processes (eg, diagnostic capacity, referrals, access to specialist care, ability to prevent and treat side effects). GCC 5-year survival rates are projected to improve from 37% to ≥50% with expanded treatment access or improved service delivery and to 42% with improved access to chemotherapy only.[13] Equitable access to appropriate medicines is thus a necessary but not sufficient component of a health system equipped to treat children with cancers.

## CONCLUSIONS

The survival disparities among children with cancer between high-income and low-income and middle-income countries have been recognised as an unacceptable injustice that demands redress. Medicines are an important component in the set of interventions needed to bridge the gaps in global 5-year net survival rates for ALL, which range from 12% in Eastern Africa, 40% in South-East Asia to 90% in Western and Northern Europe, North America, Australia and New Zealand.[13] The introduction and expansion of universal health coverage in low-income and middle-income countries provide an opportunity to save lives by including paediatric cancer medicines in benefit packages. To do so require sound, context-specific estimates of needed volumes and expected costs of paediatric cancer medications.

We developed a practical, customisable tool to transparently generate such estimates based on the needs and

circumstances of different countries. Estimates can be used for advocacy and to inform procurement processes with the aim of increasing efficiency. Ultimately, evidence-based estimates of needs, together with market analysis, can help countries move from passive purchasing based on historical data to strategic procurement and proactive mitigation of potential medicine shortages.

**Author affiliations**
[1]Center of Pharmaceutical Outcomes Research, Department of Pharmacy Practice, Faculty of Pharmaceutical Sciences, Naresuan University, Tha Pho, Phitsanulok, Thailand
[2]Department of Population Medicine, Division of Health Policy and Insurance Research, Harvard Medical School and Harvard Pilgrim Health Care Institute, Boston, Massachusetts, USA
[3]Unit for Policy and Economic Research in Childhood Cancer, Centre for Global Child Health, The Hospital for Sick Children, Toronto, Ontario, Canada
[4]Division of Haematology/Oncology, Department of Paediatrics, The Hospital for Sick Children, Toronto, Ontario, Canada
[5]Child Health Evaluative Sciences, Peter Gilgan Centre for Research and Learning, The Hospital for Sick Children, Toronto, Ontario, Canada
[6]Institute of Health Policy, Management and Evaluation, University of Toronto, Toronto, Ontario, Canada
[7]Dana-Farber/Boston Children's Cancer and Blood Disorders Center, Boston, Massachusetts, USA
[8]Center for Health Decision Science, Harvard University T H Chan School of Public Health, Boston, Massachusetts, USA
[9]Department of Pediatrics, Harvard Medical School, Boston, Massachusetts, USA
[10]Division of General Pediatrics, Boston Children's Hospital, Boston, Massachusetts, United States

**Acknowledgements** This work would not have been possible without the numerous contributions from Dr Peter Stephens from IQVIA, UK. We gratefully acknowledge Dr Stephens and IQVIA for providing the sales data and information on medicines sales data. We gratefully acknowledge the contribution of Prof Dr Suradej Hongeng, Department of Pediatrics, Faculty of Medicine, Ramathibodi Hospital, Mahidol University, Bangkok, Thailand in providing important information on the treatment of pediatric ALL in Thailand. We gratefully acknowledge the Global Task Force on Expanded Access to Cancer Care and Control, led by Dr Felicia Knaul, which provided funding for the initial development in 2014 of the approach to estimate gaps between needs for and availability of medications to treat children with cancers.

**Contributors** RS and AF contributed equally to this paper. RS, DR-D and AKW with contribution from ALF developed a tool to estimate need and availability as part of the POEMS project in 2014. The tool was updated and enhanced between 2019 and 2020 by AF and RS who also compiled the data and conducted the analysis. SG, AD and ALF advised on clinical aspects. JMY and ZJW advised on analytical aspects. AF wrote the first draft of the paper; all authors contributed to revisions of the manuscript to its final version and approved the final version.

**Funding** The initial development of a tool (POEMS project 2014) on which this work built was funded by the Global Task Force for Cancer Care and Control (grant number N/A), an initiative promoted by the Harvard Global Equity Initiative, Harvard Medical School, Harvard T.H. Chan School of Public Health, Dana-Farber Cancer Institute, Fred Hutchinson Cancer Research Center, and University of Washington School of Medicine. At that time, RS was a research fellow in the Department of Population Medicine and supported by the Thai Royal Golden Jubilee PhD program (grant number PHD/0127/2552). There were no external funds provided for the present analyses. AF was supported by a postdoctoral Fellowship from the Swiss National Science Foundation (grant number P400PM_183877) and AKW by a Department of Population Medicine Ebert Award (grant number N/A), Harvard Medical School and Harvard Pilgrim Health Care Institute. The funders of this study had no role in study design, collection, analysis, or interpretation of the data, or writing of the report.

**Competing interests** RS is currently employed by Takeda (Thailand) Ltd. AF reports personal fees from the European Society of Medical Oncology, separate from the submitted work. AKW reports support from the Global Task Force for Cancer Care and Control for the initial development of the tool.

**Patient consent for publication** Not required.

**Ethics approval** This study was approved by the Harvard Pilgrim Health Care Institute Institutional Review Board.

**Provenance and peer review** Not commissioned; externally peer reviewed.

**Data availability statement** All data relevant to the study are included in the article or uploaded as supplementary information.

**ORCID iDs**
Rosarin Sruamsiri http://orcid.org/0000-0002-1437-2492
Alessandra Ferrario http://orcid.org/0000-0002-1572-3581
Dennis Ross-Degnan http://orcid.org/0000-0002-0066-1242
Avram E Denburg http://orcid.org/0000-0003-0039-0742
A Lindsay Frazier http://orcid.org/0000-0003-4748-8552
Sumit Gupta http://orcid.org/0000-0003-1334-3670
Zachary J Ward http://orcid.org/0000-0002-4007-2207
Jennifer M Yeh http://orcid.org/0000-0002-2724-7404
Anita Katharina Wagner http://orcid.org/0000-0002-1069-6668

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
