## [Reviewer comments · BMJ Open]

ARTICLE DETAILS

TITLE (PROVISIONAL)	What are the volume and budget needs to provide chemotherapy to all children with acute lymphoblastic leukemia in Thailand? Development and application of an estimation tool
AUTHORS	Sruamsiri, Rosarin; Ferrario, Alessandra; Ross-Degnan, Dennis; Denburg, Avram; Frazier, A. Lindsay; Gupta, Sumit; Ward, Zachary; Yeh, Jennifer; Wagner, Anita Katharina

VERSION 1 – REVIEW

REVIEWER	Andrew Lofts Gray University of KwaZulu-Natal, South Africa
REVIEW RETURNED	13-Jul-2020

GENERAL COMMENTS	The authors have provided a well-characterised model for estimating both the budget impact of a particular condition/disease pairing in a paediatric population, and for considering, on the basis of existing sales data, whether access appears to be adequate in a particular setting. The model can be applied more widely, in conditions where a clear list of medicines is available (e.g. from an EML), and where the majority of use can be linked to a particular diagnosis, and finally, where reliable estimates of disease burden are available. While focused specifically on paediatric cancer care, the model can also be applied in other disease areas, provided the necessary input data are available. The limitations of the data and the methods proposed have been adequately identified, and bolstered by the use of appropriate sensitivity analyses. I have just two very minor suggestions for Table 3:  1. The "IT" route of administration is not identified in the legend notes. 2. the INN tioguanine is appropriately used, but it may be useful to include a note that this is also known as thioguanine (e.g. in USAN terms). Lastly, it might be useful to add a short comment dealing with how
--

REVIEWER	Sonya Cressman Simon Fraser University, Canada
REVIEW RETURNED	23-Jul-2020

GENERAL COMMENTS	This study uses price data from Thailand's national pharmacies and incidence data from cancer registries to calculate the cost of chemotherapy drugs to treat pediatric ALL and speculate of volume needed in this population with a microsimulation model that uses their cost analysis. Their findings put expenditure into
---

	perspective by expressing the volume needed for pediatric ALL as a ratio of the total volume used, is a subgroup analysis of drugs specifically indicated for ped ALL. The study is important and imbues a population with disparity in outcomes and one that is often underrepresented in clinical trials. The authors have the opportunity to develop global standards with their method that can be used for international comparisons. A global standard could be a valuable policy outcome that would be worth expanding on or strengthening their arguments for which could also increase the impact of the paper. It is written well with sound and clearly described methods. The article would benefit from some minor revisions such as those suggested below: Abstract The objective statement could explain why we are asking the question about drug costs for pediatric ALL. As written, it doesn't accurately convey the reason for doing the study. Abstract reads specific to Thailand only, when what I think the authors intend is to develop a toolkit or costing method for which all countries can benchmark medicine accessibility with standardizing cost and incidence rate comparisons. Strengths and limitations of this study Suggest adding the potential for developing global standards here. Background Lines 8-9: add how much of this is due to access to essential medicines (versus standard care or transplant.) Need to introduce the rationale behind volume needed vs volume sold analysis. Methods How easy is it to get IQVIA sales data? Would cost to these data be a barrier for LMICs to do their own resource utilization research? The sensitivity analysis could include a scenario where some of the newer drugs become available. Currently there are several high-cost, on-patent inhibitors and antibody-based drugs in phase 3 trials for 0-21 year old ALL patients. These will have a substantial impact on the availability of medicines in the near future. How would this impact the result? How do the authors handle wastage from unused vials or vial splitting?
--	---

VERSION 1 – AUTHOR RESPONSE

Reviewer: 1

The authors have provided a well-characterised model for estimating both the budget impact of a particular condition/disease pairing in a paediatric population, and for considering, on the basis of existing sales data, whether access appears to be adequate in a particular setting. The model can be applied more widely, in conditions where a clear list of medicines is available (e.g. from an EML), and where the majority of use can be linked to a particular diagnosis, and finally, where reliable estimates of disease burden are available. While focused specifically on paediatric cancer care, the model can also be applied in other disease areas, provided the necessary input data are available. The limitations of the data and the methods proposed have been adequately identified, and bolstered by the use of appropriate sensitivity analyses.

- *Authors' reply: Thank you for the positive comments.*

I have just two very minor suggestions for Table 3:

1. The "IT" route of administration is not identified in the legend notes.
2. the INN tioguanine is appropriately used, but it may be useful to include a note that this is also known as thioguanine (e.g. in USAN terms).

Authors' reply: Thank you, we have added a note about 1 (IT: intrathecal) and 2 in the legend of Table 3.

Lastly, it might be useful to add a short comment dealing with how

Authors' reply: This sentence is incomplete, so we are not able to address this comment.

Reviewer: 2

This study uses price data from Thailand's national pharmacies and incidence data from cancer registries to calculate the cost of chemotherapy drugs to treat pediatric ALL and speculate of volume needed in this population with a microsimulation model that uses their cost analysis. Their findings put expenditure into perspective by expressing the volume needed for pediatric ALL as a ratio of the total volume used, is a subgroup analysis of drugs specifically indicated for ped ALL. The study is important and imbues a population with disparity in outcomes and one that is often underrepresented in clinical trials. The authors have the opportunity to develop global standards with their method that can be used for international comparisons. A global standard could be a valuable policy outcome that would be worth expanding on or strengthening their arguments for which could also increase the impact of the paper. It is written well with sound and clearly described methods.

- *Authors' reply: Thank you for the positive comments.*

The article would benefit from some minor revisions such as those suggested below:

Abstract

The objective statement could explain why we are asking the question about drug costs for pediatric ALL. As written, it doesn't accurately convey the reason for doing the study.

Abstract reads specific to Thailand only, when what I think the authors intend is to develop a toolkit or costing method for which all countries can benchmark medicine accessibility with standardizing cost and incidence rate comparisons.

Authors' reply: Thank you for pointing this out, we have tried to make this clearer to the extent that the word count limits allow. We have added: "Insufficient access to anti-cancer medicines may contribute to the wide survival differences of children with cancers across the globe. We developed a tool to estimate the volume of medicines and budget requirements to provide chemotherapy to children with acute lymphoblastic leukemia (ALL)."

Strengths and limitations of this study

Suggest adding the potential for developing global standards here.

Authors' reply: We agree that this tool can be used across diseases, settings, and time. We clarified in the 'Strengths and limitations' section that the tool is dynamic and facilitates transparency for estimating the volume of medicines and budget needs. It allows users to input local data of disease incidence, treatment protocols, and medicines' prices.

Background

- Lines 8-9: add how much of this is due to access to essential medicines (versus standard care or transplant.)

Authors' reply: We assume that this refers to the survival gap. We agree that contributions of different diagnostic and treatment technologies to the survival gap are important to know. Since those contributions depend on the type cancer, type of health care setting, and synergies between diagnostic, treatment, supportive, and follow-up care interventions it would be challenging to isolate effects of individual modalities.

- Need to introduce the rationale behind volume needed vs volume sold analysis.

Authors' reply: Thank you, we have made this clearer in the last paragraph of the introduction (additions in bold)

"Together with other health system barriers, insufficient access to anti-cancer medicines may contribute to the wide survival differences of children with cancers across the globe. The objective of our study was to develop a tool to estimate medicine needs and budgetary requirements for treating children with acute lymphoblastic leukemia (ALL) and apply it to Thailand as an example case. We populated the tool with a unique set of data including the latest available estimates of children with diagnosed and undiagnosed ALL, pharmaceutical sales data and local data on cancer stage distribution. Further, we compared needed quantities with sold quantities of pediatric cancer medicines used for the treatment of ALL to identify possible unmet need. Ultimately, we aimed to generate a transferable approach that can be customized to other jurisdictions and populated with local data to estimate the need for and availability of essential pediatric cancer medicines, as well as

their corresponding budget impact. **We expect this tool to be useful for procurement planning, budgeting, advocacy, and monitoring of sufficiency of medicine supplies.**”

Methods

- How easy is it to get IQVIA sales data? Would cost to these data be a barrier for LMICs to do their own resource utilization research?

Authors’ reply: IQVIA has provided data free of charge to researchers in low- and middle-income countries on various occasions. Further, IQVIA data are not the only sources of sales data. Countries can use their own procurement data (currently underutilized) for such analyses.

- The **sensitivity analysis** could include a scenario where some of the newer drugs become available. Currently there are several high-cost, on-patent inhibitors and antibody-based drugs in phase 3 trials for 0-21 year old ALL patients. These will have a substantial impact on the availability of medicines in the near future. How would this impact the result?

Authors’ reply: We appreciate this comment as it underscores the relevance and flexibility of the tool. New drugs will be part of different treatment regimens for subsets of patients. The tool we developed is fully transparent and easily customizable to allow for addition of new treatment regimens for the subset of patients for which these will be indicated once the new drugs become available and part of clinical guidelines (and insurance benefits).

- How do the authors handle wastage from unused vials or vial splitting?

Authors’ reply: The amount wasted depends on the extent to which it is possible to share vials on a given clinic day (which depends on numbers of patients seen) and the amount required per patient which depends on treatment stage and body surface area. We added two scenarios to our sensitivity analysis. If 20% of the vial content (for all medicines that are used in vial form) is wasted, an additional US\$ 123,439 would be required to treat all diagnosed children in 2017. The amount increases to US\$ 246,876 if 40% of the volume of each vial is wasted. We made the necessary additions to the manuscript.

VERSION 2 – REVIEW

REVIEWER	Sonya Cressman Simon Fraser University, Canada
REVIEW RETURNED	09-Sep-2020

GENERAL COMMENTS	The authors have responded to minor suggestions from two positive reviewers and the editor's requests. They thoughtfully now include a sensitivity analysis to address minor concerns raised in the initial round of peer review. The scope of the tool they used in this study has been expressed as a potential for application in other LMIC settings. The work is clearer and more accurately conveys the impact. Thank you for inviting me to review its revision.
---